# Inflammatory Predictors of Prognosis in Patients with Traumatic Cerebral Haemorrhage: Retrospective Study

**DOI:** 10.3390/jcm11030705

**Published:** 2022-01-28

**Authors:** Piotr Defort, Natalia Retkowska-Tomaszewska, Marcin Kot, Paweł Jarmużek, Anna Tylutka, Agnieszka Zembron-Lacny

**Affiliations:** 1Neurosurgery Center University Hospital, Collegium Medicum University of Zielona Gora, 28 Zyty Str., 65-417 Zielona Gora, Poland; natalia.retkowska@gmail.com (N.R.-T.); m.kot@cm.uz.zgora.pl (M.K.); p.jarmuzek@cm.uz.zgora.pl (P.J.); 2Department of Applied and Clinical Physiology, Collegium Medicum University of Zielona Gora, 28 Zyty Str., 65-417 Zielona Gora, Poland; a.tylutka@cm.uz.zgora.pl (A.T.); a.zembron-lacny@cm.uz.zgora.pl (A.Z.-L.)

**Keywords:** brain injury, falls, lymphocytes, neutrophils, systemic inflammation

## Abstract

We aimed to evaluate the relationship between neutrophil to lymphocyte ratio (NLR), platelet to lymphocyte ratio (PLR), lymphocyte to monocyte ratio (LMR), systemic inflammation index (SII), and Glasgow Coma Scale (GCS) score in patients with traumatic intracerebral haemorrhage (TICH). We retrospectively investigated 95 patients with TICH hospitalised at the Neurosurgery Department in Zielona Gora from January 2017 to March 2021. Routine blood tests were performed 5 h after injury. NRL and SII were significantly higher in patients with GCS ≤ 8 than patients with GCS > 8 and exceeded reference values in 95% of patients. GCS was inversely correlated with NLR and SII. Receiver operating characteristic (ROC) analysis confirmed the value of NLR and SII regarding GCS score; Area Under the Curve (AUC) 0.748, 95% Confidence Interval (CI) 0.615–0.880. An optimised NLR cut-off value of 0.154 was identified with a sensitivity of 0.90 and specificity of 0.56. The value of SII regarding GCS was confirmed with ROC curves; AUC 0.816, 95% CI 0.696–0.935. An optimised NLR cut-off value of 0.118 was identified with a sensitivity of 0.95 and specificity of 0.57. NLR and SII are significantly related to GCS scores and are promising predictors of clinical prognosis in TICH patients.

## 1. Introduction

Traumatic brain injury (TBI) remains a major cause of death and disability worldwide. For all ages and TBI severities, crude incidence rates ranged from 47.3 to 694 per 100,000 population per year (country-level studies) and from 83.3 to 849 per 100,000 population per year (regional-level studies). Mortality rates ranged from 9 to 28.10 per 100,000 population per year (country-level studies) and from 3.3 to 24.4 per 100,000 population per year (regional-level studies). The most common mechanisms of injury were traffic accidents and falls [1]. Patients with TBI are most commonly categorised into mild, moderate, and severe based on Glasgow Coma Scale (GCS) [2], as well as findings from computed tomography (CT) for the head [3]. Magnetic resonance imaging (MRI) is routinely performed at a growing number of centres and is particularly helpful in the quantification of diffuse axonal TBI. The more recently developed diffusion tension DT-MRI can detect in even greater detail alterations in the microstructure of the white matter, and it shows considerable promise in the assessment of neuronal damage [4]. However, these neuroimaging techniques reveal little or no information regarding secondary injury processes such as excitotoxicity, neuroinflammation, blood–brain barrier breakdown, ischemic damage, and cell death. New biomarkers are a promising tool for clinicians for obtaining additional, patient-specific information.

A substantial subgroup of patients with TBI admitted to hospitals includes patients with traumatic intracerebral haemorrhage (TICH). In these cases, damage to the neuronal structure is apparent on admission CTs. The earliest consequences of TICH are increased vascular permeability, leading to brain oedema and complex inflammatory response [5,6,7,8]. Alterations in blood–brain barrier permeability may contribute to secondary brain injury through an abnormal passage of blood-borne substances into the extravascular space, increasing neuronal vulnerability [9,10]. Increased vascular permeability is associated with multiple post-traumatic inflammatory events [9,11]. Trauma-induced mechanical damage to vascular cell membranes results in endothelial cell necrosis and red blood cell extravasation [12,13]. Endothelial and leukocyte adhesion molecules are upregulated as a result of primary traumatic events, leading to the recruitment and extravasation of circulating leukocytes into the brain parenchyma [14]. Blood-borne circulating monocytes gain access to the brain parenchyma and may contribute to macrophage-induced secondary injury mechanisms. Moreover, a brain injury is often accompanied by multi-organ trauma, which is often severe and difficult to diagnose and treat and may influence prognosis [15].

The prognosis of patients with TICH is still unclear, and multiple studies have tried to identify predictors of outcome in these patients [9]. An assessment of the level of consciousness with the Glasgow Coma Scale at admission remains the most reliable prognostic factor, which unfortunately can be subject to human error. It may also be underestimated due to drug or alcohol intoxication [16]. Thus, the need for more objective prognostic factors should be addressed. The immunological studies demonstrate diagnostic potential and provide information regarding both the acute as well as long-term cerebrovascular consequences of brain injury leading to mortality, disability, and cognitive impairment [17]. In the past few years, many studies reported that the combinations of the haematological components of the systemic inflammatory responses, such as the neutrophil-to-lymphocyte ratio (NLR), platelet-to-lymphocyte ratio (PLR), the lymphocyte-to-monocyte ratio (LMR), and systemic immune inflammation index (SII) were effective prognostic indicators in patients with a severe traumatic brain injury [10,17,18], neoplasms [19,20,21,22,23], obesity [24,25], coronary artery disease [26,27], diabetes [28], and acute ischaemic stroke [29]. The components of these easily calculated parameters are readily available, inexpensive, and routinely measured in daily practice as part of the complete blood count [30]. The calculation of these haematological components of the systemic inflammatory response may provide clinicians with a further valuable and objective tool that would support the clinical assessment of the head trauma patients in their management and prognosis. Therefore, the study was designed to assess the diagnostic suitability of neutrophil-to-lymphocyte ratio, platelet-to-lymphocyte ratio, lymphocyte-to-monocyte ratio, and systemic immune inflammation index compared to the severity of the brain injury according to the Glasgow Coma Scale in patients with traumatic cerebral haemorrhage.

## 2. Materials and Methods

### 2.1. Patients

This retrospective study was based on a database including 198 cases of traumatic injury brain at Neurosurgery Centre University Hospital in Zielona Gora from January 2017 to March 2021. The average time from head injury to admission ranged from 1 to 12 h. The diagnosis was based on a history of injury, clinical manifestations, and radiological findings. The inclusion criteria were as follows: traumatic intracerebral haemorrhage (TICH) and age ≥18 years. Patients admitted later than 12 h after the injury and patients with incomplete and missing data were excluded from the study. Eventually, 95 patients (females *n* = 16, males *n* = 79) were included in the project (Table 1). The study patients were allocated into two groups based on Glasgow Coma Scale (GCS) score: GCS > 8 group including patients with minimal, mild, and moderate head injuries (*n* = 75) and GCS ≤ 8 group including patients with severe and critical head injuries (*n* = 20) according to the classification described by Carney et al. [31] and Maas et al. [32]. Data regarding age, gender, haematological variables, injury mechanism, GCS score, indications for operative treatment, and accompanying injuries, diseases, and addictions were collected from electronic medical records at University Hospital in Zielona Gora. Patients with traumatic cerebral haemorrhage were diagnosed using head computed tomography (CT) at admission. The study protocol was approved by the Bioethics Commission at Regional Medical Chamber Zielona Gora, Poland (No. 02/131/2020), in accordance with the Helsinki Declaration.

### 2.2. Treatments

The treatment of patients with traumatic cerebral haemorrhage in Neurosurgery Center in Zielona Gora depends on TICH severity and findings from the admission CT and is managed according to Guidelines for the Management of Severe TBI [33,34]. Tracheal intubation and mechanical ventilation were adopted based on the patients’ level of consciousness (GCS ≤ 8) and arterial oxygen saturation. If a severe mass effect caused by enlargement of TICH or intracerebral hematoma developed during hospitalisation, a decompressive craniectomy and/or evacuation of mass lesions was performed.

### 2.3. Haematological Variables and White Blood Cells

Blood sampling for laboratory tests using S-Monovette-EDTA K2 tubes (Sarstedt AG & Co. KG, Nümbrecht, Germany) was limited to the timespan within 24 h after the onset of brain injury. Haematological parameters including total white cell count (WBC), red blood cell count (RBC), platelet count (PLT), differential white cell count (neutrophils, lymphocytes, monocytes, eosinophils, and basophils), and haemoglobin (HB) were determined by Sysmex XN-1000 (Europe Gmbh, Norderstedt, Germany). NLR, PLR, LMR, and SII were calculated and compared to references values according to Luo et al. [30]. The formulas are as follows: NLR = Neutrophil count (10^3^/µL)/Lymphocyte count (10^3^/µL); PLR = Platelet count (10^3^/µL)/Lymphocyte count (10^3^/µL); LMR = Lymphocyte count (10^3^/µL)/Monocyte count (10^3^/µL); SII = (Platelet count (10^3^/µL) × (Neutrophil count (10^3^/µL)/Lymphocyte count (10^3^/µL)).

### 2.4. Statistical Analysis

Continuous variables were reported as mean values ± standard deviation (SD) and median with interquartile range (25–75%), while categorical variables were expressed as count and percentage. The statistical significance of intergroup differences (according to GCS) was compared through the Kruskal–Wallis non-parametric test and through Pearson’s χ^2^ test for categorical variables. Receiver operating characteristic (ROC) curves showed sensitivity vs. specificity such that the area under the curve (AUC) varied from 0.5 to 1.0, with increased values demonstrating higher discriminatory ability. Univariate logistic regression analyses were performed to separately examine the association between unfavourable outcomes and each of the indicators. Correlations between variables and GCS score were analysed through Spearman correlation tests (R Spearman rank correlation coefficient). The value *p* < 0.05 was considered to represent a statistically significant difference. All analyses were performed by using the R version 4.0.3 [35].

## 3. Results

### 3.1. Patients

A total of 95 patients were included in this study, with 83% being males; the median age of patients was 51 years (Table 1). TICH was mainly caused by falls (42% of all patients: 43.5% in GCS > 8 group and 30% in GCS ≤ 8 group). The traffic accidents were the cause of TICH in 19% of patients; most of them (63%) had GCS ≤ 8 at admission. This group of patients constituted 55% of all patients in GCS ≤ 8 group. The seizures and violence were also important causes of TBI in GCS > 8 group of the studied patients. There were no patients in GCS ≤ 8 group with TBI caused by seizures. The other or unknown causes of TICH accounted for 19%; most of these patients were allocated to GCS > 8 group of the study (89%). The patients with severe and critical head injuries (GCS ≤ 8 group) were significantly (*p* < 0.01) younger than patients with minimal, mild, and moderate head injuries. Thirty-seven per cent of all patients with traumatic cerebral haemorrhage (TICH) had suffered the injury under the influence of alcohol. Fifteen per cent of patients were operated on during the first 24 h after the cerebral injury, 9.3% (*n* = 7) of patients in GCS > 8 group, and 35% (*n* = 7) of patients in GCS ≤ 8 group. There were three cases of deaths during hospitalisation; two of them were classified as brain death and subjected to the organ transplant procedure.

### 3.2. Haematological Variables and White Blood Cells

The haematological markers were within the referential ranges, and they did not significantly differ between groups (Table 2). However, glucose concentration was significantly higher in patients with GCS ≤ 8 but it was not related to causes of TICH. The counts of WBC, neutrophils, and monocytes were above reference values in both groups, whereby a 2-fold increase was recorded in patients with ≤8 GCS (Table 3). Similarly, NRL and SII were significantly higher in patients with GCS ≤ 8 and exceeded reference values in 95% of patients. GCS was inversely correlated with NLR and SII (Table 4). There was no significant correlation between GCS score and PLR and LMR counts. ROC analysis confirmed the value of NLR regarding GCS score (AUC 0.748, 95% confidence interval CI, 0.615–0.880, *p* < 0.001) (Table 5). An optimised NLR cut-off value (Youden Index) of 0.154 was identified with a sensitivity of 0.90 and specificity of 0.56. The value of SII regarding GCS score was also confirmed with ROC curves (AUC 0.816; 95% CI; 0.696–0.935, *p* < 0.0001). An optimised NLR cut-off value of 0.118 was identified with a sensitivity of 0.95 and specificity of 0.57 (Figure 1 and Figure 2).

## 4. Discussion

Traumatic brain injury, particularly TICH, is one of the most complex and diverse pathological medical conditions. According to the epidemiological investigation in Poland in 2009–2012 [33], the predominant external causes of TICH were traffic accidents (40%) followed by falls (33%). Our study shows the opposite trend, i.e., TICH was mainly caused by falls (42%), and traffic accidents contributed to TICH in 19% of patients. The shift from road traffic incidents to falls as the most common mechanism of injury in Europe was observed in recent studies [1]. There is also an apparent trend for increasing the percentage of falls as a cause of TICH as the age of patients advances. We observed a similar shift in our study.

Age and sex breakdown were reported in the majority of studies. However, it is difficult to compare the distribution of TICH across populations, as many studies report the data concerning only selected parts of the population (children or older adults, or adults only, etc.). According to a systemic review by Brazinova et al. [1], the lowest reported mean age was 26.7 years in the Republic of San Marino, and the highest value reached 44.5 years in Austria in 2009–2011. The reported proportion of males was always greater than that of females (irrespective of age, severity, and mechanism of injury), ranging from 55% in Sweden in 2001 to 80% in Ireland in 2005–2007. This may reflect the fact that the major causes of traumatic brain injury are related to more male-dominated activities. However, the proportion of men in TBI studies decreased with age. In our study, males constituted over 80% of patients, and they were significantly younger, i.e., 37.8 ± 19.2 years for GCS ≤ 8 and 53.9 ± 18.6 years for GCS > 8 group. The difference persisted when the whole population was taken into account. This might be due to a significantly higher percentage of falls as a cause of head trauma in the older population in comparison to high-energy traffic accidents, which dominate in younger patients.

In the Polish study by Miekisiak et al. [36], the most common diagnosis was TICH (82 × 10^5^ per year), and the most prevalent injury requiring surgery was subdural haematomas (15 × 10^5^ per year). In our study, patients diagnosed with subdural and intracerebral haematoma at admission were excluded from the study; however, there were 13 patients (14%) requiring evacuation of intracerebral haematomas, which developed 12 h after the admission. The prognosis in patients who have not required surgical treatment and those who have undergone surgery is still unclear, and multiple studies have tried to identify predictors of outcome in these patients [37]. In our group, every patient, who needed surgical intervention later in the course of treatment, was found to have high values of NLR and SII above reference levels. Fourteen patients were operated on during the first 24 h after the cerebral injury. However, due to the small sample size, no conclusion can be made for NLR and SII as prognostic factors for the development of intracerebral haematoma and/or progression of contusions, as was presented in the study by Zhuang et al. [18]. The authors demonstrated that NRL allowed identifying the risk factors for surgical intervention among patients with the growth of TICH.

The inflammatory response to brain injury is observed following ischemic, traumatic, or excitotoxic brain injury or seizure and often contributes to a long-term disability. At sites of brain injury, damaged cells release the factors that trigger the inflammatory cascade, along with chemokines and growth factors, which attract neutrophils and monocytes. The first cells attracted to the site of injury include neutrophils, followed by monocytes, lymphocytes, and mast cells. The migratory activity of neutrophils allows them to accumulate at sites of brain injury within a few hours [38]. Accordingly, neutrophils increase oxidative stress, exacerbate blood–brain barrier damage, and promote neuronal cell death. This is manifested by a poor functional outcome and an increased mortality rate in patients with various degrees of TBI. The role of T lymphocytes in TBI remains largely unknown. T cells are recruited by the reactive oxygen species released from neutrophils, but experimental data show that T cells play no significant role in the pathogenesis of early TBI [39]. In our study, the count of neutrophils was above reference values in all patients, whereas it was approx. 35% higher in GCS ≤ 8 than in GCS > 8 group. There was no difference in the lymphocyte count. Although neutrophils seem to play a pathogenic role during TBI, their advantageous role in the wound-healing process should not be neglected [40].

The changes in NRL and SII are strictly dependent on neutrophils, which represent from 50% to 70% of the total circulating leukocytes [36]. In our study, NLR and SII were significantly higher in patients with GCS ≤ 8 than GCS >8. Similarly, Chen et al. [17] observed the peak NRL in patients with severe brain injury and indicated NRL as a promising predictor for 1-year outcomes. Siwicka-Gieroba et al. [41] showed that NRL elevation predicted mortality in severe TBI patients. According to Acar et al. [42], NLR can suggest brain pathologies on CT scans of patients with isolated minor head trauma and with a GCS score of 15 when the need for a CT scan is not clear. Recently Reznik et al. [43] demonstrated that early NLR elevation also predicted delayed-onset delirium, potentially implicating systemic inflammation as a contributory delirium mechanism. There is much less research reporting the analyses of SII in brain injury. Trifan and Testai [44] concluded that SII was an independent predictor of poor outcomes at hospital discharge in patients with spontaneous intracerebral haemorrhage. We demonstrated that SII was even 2-fold higher in patients with GCS ≤ 8 including patients with severe and critical head injuries. In contrast, some authors did not confirm the sensitivity of NLR or SII in predicting poor outcomes in TBI patients, thereby suggesting that other haematological parameters have a higher sensitivity as prognostic markers [45,46]. However, most studies generally support the view of the predictive value of NLR and SII in patients with brain injury.

NLR and SII, inversely correlated with GCS, are considered to be the best method for the initial classification of TBI patients and to define the route of administration. However, ethanol exposure may confound the initial assessment of GCS, leading to the exposed patients being assessed as more severe cases. The percentage of patients being intoxicated at the time of admission ranges between 30% and 60% in different studies [16]. According to Xu et al. [47], alcohol may alleviate the TBI-induced pro-inflammatory response. In our study, 37% of all the patients with TICH were under the influence of alcohol, but their values of NLR and SII did not differ when compared to other patients. Ethanol intoxication increases the risk of traumatic brain injury, but, on the other hand, the presence of ethanol following isolated moderate–severe traumatic brain injury may be neuroprotective. Therefore, further studies are recommended to investigate the interaction between ethanol exposure and brain injury and the potential of ethanol as a therapeutic agent [16].

## 5. Conclusions

Inflammation is frequently a key element in the pathological progression of traumatic brain injury; therefore, the analysis of changes in NLR and SII as exponents of the systemic inflammatory response can be an important supplement to conventional methods in determining prognosis and possibly selecting patients requiring closer monitoring.

## 6. Limitations

The limitations of the study include a relatively small number of patients, especially the group with ≤8 points in GCS score, and no observations in the following days after TICH.

## Figures and Tables

**Figure 1 jcm-11-00705-f001:**
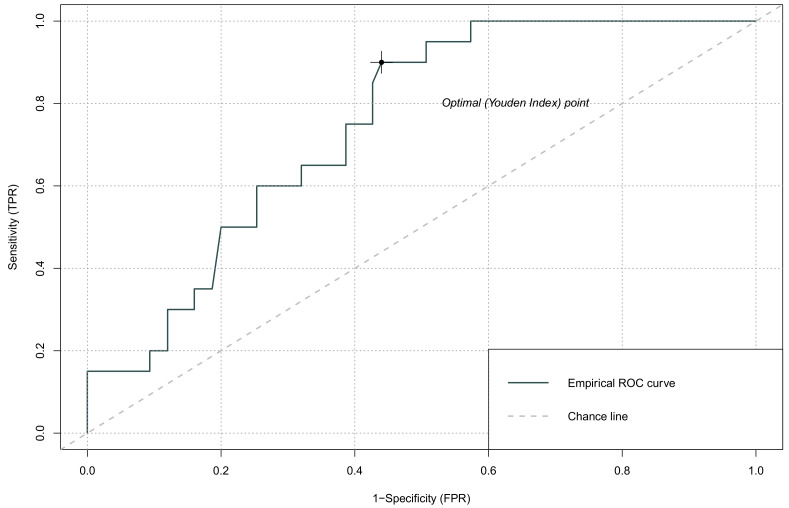
ROC curve analysis for admission GCS score and NLR; FPR—false positive rate, TPR—true positive rate.

**Figure 2 jcm-11-00705-f002:**
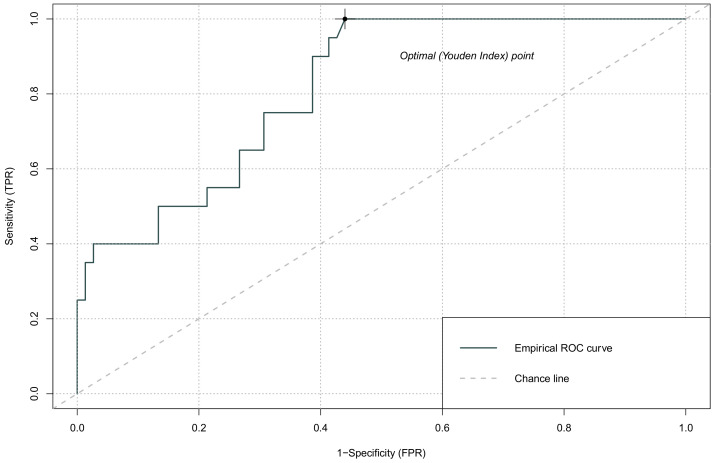
ROC curve analysis for admission GCS score and SII; FPR—false positive rate, TPR—true positive rate.

**Table 1 jcm-11-00705-t001:** The demographic and clinical data of the TICH patients.

	Patients GCS > 8	Patients GCS ≤ 8
Characteristics		
Number of subjects	75	20
Median age (years)	51 (21–100)	29 (18–85)
Mean ages (years)	53.9 ± 18.6	37.8 ± 19.2
Number of females, *n* (%)	12 (16)	4 (20)
Number of males, *n* (%)	63 (84)	16 (80)
Surgery within the first 24 h, *n* (%)	7 (9.3)	7 (35)
Injury mechanisms, *n* (%)		
Falls	34 (45.3)	6 (30)
Traffic accident	7 (9.3)	12 (55)
Seizures	12 (16)	0 (0)
Violence	6 (8)	1 (5)
Others/unknown	16 (21.3)	2 (10)

Abbreviations: GCS—Glasgow Coma Scale, TICH—traumatic cerebral haemorrhage.

**Table 2 jcm-11-00705-t002:** Haematological variables and glucose.

	Reference Values	Patients GCS > 8		Patients GCS ≤ 8		
		Mean ± SD	Med (iqr 25–75%)	Mean ± SD	Med (iqr 25–75%)	*p*-Value
RBC (10^6^/µL)	4.2–6.5	4.3 ± 0.8	4.4 (3.9–4.7)	4.3 ± 0.7	4.3 (3.8–4.8)	0.823
HB (g/dL)	12.0–18.0	13.7 ± 1.8	14.1 (12.6–14.9)	13.2 ± 1.8	13.4 (12.2–4.6)	0.308
HCT (%)	38.0–54.0	42.2 ± 10.2	42.1 (38.1–44.4)	39.0 ± 5.2	39.7 (35.8–42.7)	0.111
MCV (fL)	80.0–97.0	93.7 ± 11.5	94.8 (90.9–99.0)	89.3 ± 12.7	89.6 (86.8–96.9)	0.008
MCH (pg/RBC)	26.0–32.0	32.1 ± 2.0	31.7 (30.6–33.6)	33.8 ± 12.6	31.1 (29.2–33.0)	0.124
MCHC (g/dL)	31.0–36.0	33.4 ± 2.6	33.6 (32.9–34.4)	33.6 ± 1.6	33.2 (32.4–34.8)	0.658
RDW (%)	11.5–14.8	13.0 ± 2.1	12.2 (11.3–14.0)	13.3 ± 5.6	12.4 (11.0–14.2)	0.161
Glucose (mg/dL)	60.0–99.0	132.4 ± 40.5	124 (107–155)	164.1 ± 88.2	155 (127–195)	*p* < 0.05

Abbreviations: GCS—Glasgow Coma Scale score, SD—standard deviation, Med—median, iqr—interquartile range, RBC—red blood cells, HB—haemoglobin, HCT—haematocrit, MCV—mean cell volume, MCH—mean corpuscular haemoglobin, MCHC—mean corpuscular haemoglobin concentration, RDW—red cell distribution width.

**Table 3 jcm-11-00705-t003:** White blood cells and platelets counts.

	Reference Values	Patients GCS > 8		Patients GCS ≤ 8		
		Mean ± SD	Med (iqr 25–75%)	Mean ± SD	Med (iqr 25–75%)	*p*-Value
WBC (10^3^/µL)	4.0–10.2	12.1 ± 4.8	11.2 (8.7–14.3)	22.3 ± 3.7	16.0 (12.2–22.3)	*p* < 0.001
Neutrophils (10^3^/µL)	2.0–6.9	9.9 ± 3.1	8.7 (6.0–12.0)	13.5 ± 4.8	12.7 (9.5–17.3)	*p* < 0.01
Lymphocytes (10^3^/µL)	0.6–3.4	1.9 ± 1.2	1.6 (1.2–2.2)	3.4 ± 2.9	2.5 (1.3–2.8)	0.451
Monocytes (10^3^/µL)	0.00–0.90	0.8 ± 0.5	0.6 (0.5–0.9)	2.0 ± 0.5	0.9 (0.6–1.2)	*p* < 0.05
Platelets (10^3^/µL)	140–420	211 ± 74	207 (159–247)	226 ± 103	221 (190–247)	0.067
NLR (10^3^/µL)	0.87–4.15	5.4 ± 3.3	5.3 (3.1–6.4)	9.6 ± 6.4	8.3 (5.3–9.7)	*p* < 0.001
PLR (10^3^/µL)	47–198	119 ± 44	119 (92–144)	146 ± 71	142 (86–162)	0.340
LMR (10^3^/µL)	2.45–8.77	3.1 ± 1.9	3.1 (1.8–3.4)	2.9 ± 2.3	2.6 (1.3–3.2)	0.287
SII (10^3^/µL)	142–808	995 ± 535	977 (588–1216)	2081 ± 1244	1970 (1188–2234)	*p* < 0.0001

Abbreviations: GCS—Glasgow Coma Scale score, SD—standard deviation, Med—median, iqr—interquartile range, WBC—white blood cells, NLR—neutrophil/lymphocyte ratio, PLR—platelet/lymphocyte ratio, LMR—lymphocyte/monocyte ratio, SII—systemic immune inflammation index.

**Table 4 jcm-11-00705-t004:** Significant correlations between GCS, NLR, PLR, LMR, and SII.

	NLR	PLR	LMR	SII
GCS	R = −0.318 p < 0.0001	R = −0.047 0.621	R = 0.153 0.105	R = −0.357 p < 0.0001

R Spearman rank correlation coefficient.

**Table 5 jcm-11-00705-t005:** ROC analysis.

	AUC (95% CI)	Cut-Off	Sensitivity	Specificity
NLR	0.748 (0.615–0.880)	0.154	0.90	0.56
PLR	0.570 (0.424–0.715)	0.244	0.40	0.82
LMR	0.422 (0.285–0.559)	0.227	0.35	0.84
SII	0.816 (0.696–0.935)	0.118	0.95	0.57

Abbreviations: AUC—area under the curve, CI—confidence interval, NLR—neutrophil/lymphocyte ratio, PLR—platelet/lymphocyte ratio, LMR—lymphocyte/monocyte ratio, SII—systemic immune inflammation index.

## Data Availability

The data used to support the findings of this study are available from the corresponding author upon request.

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
