# Peer review of "Inflammatory Predictors of Prognosis in Patients with Traumatic Cerebral Haemorrhage: Retrospective Study"

_jcm, 2022, doi:10.3390/jcm11030705_

Round 1

Reviewer 1 Report

I recommend to publish the article

Reviewer 2 Report

Authors have modified the manuscript according to the comments.

I suggest to include the ethical statement in the manuscript.

This manuscript is a resubmission of an earlier submission. The following is a list of the peer review reports and author responses from that submission.

Round 1

Reviewer 1 Report

Despite being a relevant topic, the article “Inflammatory predictors of prognosis in patients with traumatic cerebral haemorrhage; retrospective study” by Retkowska-Tomaszewska et al., does not stand out from other published studies. In regards to this, the small number of subjects together with a simplistic and/or limited number of clinical characteristics and laboratory parameters analyzed (among a multitude that can be find in the literature) pose more questions than solid results. The novelty part is somewhat vague and some key and recent studies related to this topic are not cited. 

Reviewer 2 Report

Inflammatory predictors of prognosis in patients with traumatic cerebral hemorrhage; retrospective study

The aim of the study was to evaluate the relationship between different standard blood parameter ratios and Glasgow Coma Scale (GCS) scores in patients with traumatic brain injury.

This is an interesting study. I have only few comments. A well done study! Congratulations!

Row 95:           Why have you chosen GCS < 8 and GCS > 8 ? Is it an arbitrary value or what? Please, argue. This choice might influence the statistics by giving optimal statistic results. Therefore, argue either in the Method section or in the Discussion.

Presentation of results are clear, and the statistics are as far I can see appropriate.

Row 234:         Open with your main result: Move Row 275-292 up to Row 234. Then you can continue with the text from Row 234. Thus, you first present your main result. Then you can put in an ‘However’ and present ‘reservations’ like you have found more fall accidents than traffic accidents and the other differences.

A weakness of the study you might mention is that it is retrospective. However, I do not find any obvious problems.

Reference 7) Incomplete. Which year?

Reviewer 3 Report

Thank you for the opportunity to review this important paper. This paper was a very useful study on the prognosis of traumatic brain injury based on blood test data. However, I consider the following problems in printing the paper.

・It is difficult to conclude from the results of this study because the sample size is not enough.

・Although the patients were divided into two groups by GCS, there was a significant difference in age. In addition, the percentage of causes of traumatic injury is also very different. In the discussion section, the difference in the characteristics between the participants in the previous study and the participants in this study is discussed, but I think what the reader is interested in is the result when those background factors are considered.

・In L65-67, I was curious about the use of GCS as a prognostic indicator, even though the authors pointed out human error in GCS.

Reviewer 4 Report

Title: Inflammatory predictors of prognosis in patients with traumatic cerebral haemorrhage; retrospective study

Manuscript ID jcm-1533246

Manuscript Type: Original Article

I am thankful to the journal for providing me the opportunity to review the article. The study is informative.

Here are my few concerns to the author regarding manuscript-

  1. Authors did not provide any ethical statement for study.

  1. The only limitation of this study is small number of subjects.

  1. The necessity and innovation of the article should be presented to the introduction.

  1. It is suggested to rewrite conclusion part in more crisp way. This section should present in one 250-300 words paragraph.

  1. Authors should also need to proceed for dynamical study in future path.

  1. There are lot of punctuation and typographical errors throughout in the manuscript. It must be rechecked by native English speaker.

  1. Author must be provide good high resolution/quality of picture.

  1. Match all the cited references in the text part.

I strongly recommend the Original article for publications in the reputed journal after minor revision.